# A Comparison of the Sensing Behavior for Pt-Mo/C-, Pt-Zr/C-, Pt-Fe-Ir/C-, and Pt/C-Modified Glassy Carbon Electrodes for the Oxidation of Ascorbic Acid and Dopamine

Yu-Ching Weng *, Jia-Yi Su-Chen, Ting-Yu Yang and Chieh-Lin Chiang

Department of Chemical Engineering, Feng Chia University, Taichung 40724, Taiwan
* Correspondence: ycweng@fcu.edu.tw

**Abstract:** This study compares the sensing performance for platinum-molybdenum-, platinum-zirconium-, platinum-iron-iridium-, and platinum-modified electrodes in terms of the amperometric detection of ascorbic acid (AA) and dopamine (DA). The Pt, Pt-Mo, Pt-Zr, and Pt-Fe-Ir electrocatalysts are fabricated by chemical reduction on a carbon black support (XC-72) and are further modified on a glassy carbon electrode (GCE) as sensing electrodes. The Pt-Mo/C/GCE exhibits better electrocatalytic activity toward AA and DA than the Pt/C/GCE, Pt-Zr/C/GCE, and Pt-Fe-Ir/C/GCE. The Pt-Mo/C/GCE exhibits a sensitivity of 31.29 $\mu$A mM$^{-1}$ to AA at 0.3 V vs. Ag/AgCl and a sensitivity of 72.24 $\mu$A mM$^{-1}$ to DA at 0.6 V vs. Ag/AgCl and is reproducible and stable. This electrode has a respective limit of detection of 7.69 and 6.14 $\mu$M for AA and DA. Sucrose, citric acid, tartaric acid, and uric acid do not interfere with AA and DA detection. The diffusion coefficient and kinetic parameters, such as the catalytic rate constant and the heterogeneous rate constant for AA and DA, are determined using electrochemical approaches.

**Keywords:** Pt/C; Pt-Mo/C; Pt-Zr/C; Pt-Fe-Ir/C; ascorbic acid; dopamine; sensor



## 1. Introduction

Vitamin C, which is also known as ascorbic acid (AA), is an essential nutrient for human beings that exists in many vegetables and fruits. AA is necessary for the growth, development and repair of body tissue and is involved in collagen formation, iron absorption, immune system functioning, cartilage, bone and tooth maintenance, and wound healing promotion [1]. An adequate intake of AA can prevent and treat scurvy [2].

AA is also a natural antioxidant that reduces free radical damage to DNA. The accumulation of free radicals over years can initiate the aging process and creates many threats to health. Low levels of AA are associated with a risk of high blood pressure, gallbladder disease, stroke, cancer, and atherosclerosis [3]. When AA intake in excess of normal doses overloads the body, it begins to accumulate, leading to overdose symptoms [4]. The control of AA content in the blood or urine is important for clinical diagnoses; therefore, a convenient method for the accurate detection of AA concentration in biological fluids is required.

Dopamine (DA) belongs to the family of catecholamines and is an important neurotransmitter that is responsible for the transmission of desire, feeling, excitement, and happiness in the brain [5]. It is synthesized in the human brain and kidneys and plays an important role in the function of the central nervous, renal, hormonal, and cardiovascular systems [6]. Insufficient or dysregulated dopamine can lead to loss of muscle control, an inability to concentrate, and in severe cases, Parkinson's disease [7]. If there is too much dopamine in the body, the patient's limbs and trunk twitch involuntarily and Huntington's disease ensues [8]. Therefore, an accurate, sensitive, and simple assay to detect and quantify dopamine is of great importance for biomedical chemistry, neurochemistry, and diagnostic research.

High performance liquid chromatography, chemiluminescence, UV–Vis spectroscopy, capillary electrophoresis, and electroanalytical methods [9] are used to detect AA and DA.

Electrochemical detection methods have been extensively studied in recent years because of their simplicity, high sensitivity, fast response time, uncomplicated operating procedures, and low cost [10,11]. However, using traditional bare electrode materials to detect AA and DA results in low reproducibility due to electrode fouling and poor selectivity due to overlapping voltammetric peaks [12]. Therefore, the selection of appropriate electrode materials to separate the oxidation potential range of AA and DA is important.

Carbon-based materials and metal nanoparticles, such as gold and platinum, are used to modify electrodes to increase sensitivity and selectivity for the detection of AA and DA [13–16]. Gold nanoparticles (NPs) and platinum NPs have been widely used to enhance electrochemical signals because of their high electrical conductivity and electrocatalytic properties [17]. Many Pt NPs hybrids that are supported by carbon materials have been used to construct electrochemical sensors for AA and DA. Xu et al. reported a Pt/reduced graphene oxide (Pt/RGO) hybrid modified glassy carbon electrode (GCE) for the detection of DA and uric acid in the presence of high concentrations of AA [13]. Yogeswaran et al. fabricated a multi-walled carbon nanotube (MWCNT)/Nafion/PtAu composite for the simultaneous detection of AA, epinephrine, and UA [14].

Platinum is recognized as a good electrocatalyst for oxidation or reduction of small molecules. It is widely used in electrochemical sensors to detect hydrogen peroxide, glucose, ethanol, etc. [18,19]. In addition, the platinum catalyst is also very stable and would not be corroded by strong acid or alkali solution [19]. Even though platinum is the great electrocatalyst, its broad application is limited due to its expensive cost [20]. Alloying Pt with other metals to lower the Pt amounts is one of the strategies to address this issue. Thus, for this study, we chose a platinum-based electrocatalyst to detect AA and DA.

Noble metals with carbon nanostructures, including CuAg/graphite [15], AuNPs-RGO [21], AgNPs-RGO [22], graphene/Pt [23], and AgAu-MWCNTs [16], perform well for the electrochemical detection of AA and DA. A combination of nanometals and nanocarbon supports performs better than conventional counterparts because surface poisoning and agglomeration of NPs is inhibited [24]. Few studies concern the application of bimetallic or ternary metals on nanocarbon supports in sensors and electrocatalysis. To the authors' best knowledge, the synthesis of Pt-Mo/C, Pt-Zr/C or Pt-Fe-Ir/C and the electrochemical sensing applications for AA and DA have not been demonstrated.

This study compares the sensing performance for Pt/C/GCE, Pt-Mo/C/GCE, Pt-Zr/C/GCE, and Pt-Fe-Ir/C/GCE for AA and DA detection. These Pt-based electrocatalysts are prepared on a carbon black support (XC-72) by chemical reduction and modified on glassy carbon electrodes. High-resolution transmission electron microscope (HR-TEM), an energy dispersive spectrometer (EDS) and a high-resolution X-ray diffractometer (XRD) are respectively used to determine the particle size, the elemental composition, and crystallinity of the Pt-based electrocatalysts. Cyclic voltammetry and polarization curves are used to determine the reaction potential and mass transfer control potential for a Pt-based electrocatalyst for the oxidation of AA and DA. The sensitivity, response time, selectivity, and stability of these Pt-based electrocatalysts toward AA and DA are determined. A Pt-Mo/C/GCE exhibits the best sensing performance for AA and DA.

## 2. Results and Discussion

### 2.1. Characterization of the Pt/C, Pt-Mo/C, Pt-Zr/C and Pt-Fe-Ir/C Electrocatalysts

The morphology of the Pt/C, Pt-Mo/C, Pt-Zr/C, and Pt-Fe-Ir/C electrocatalysts was characterized using TEM and the results are shown in Figure 1. The Pt, Pt-Mo, and Pt-Fe-Ir electrocatalysts were uniformly dispersed on the carbon black support and the Pt-Zr electrocatalyst exhibited some obvious agglomeration. The average respective particle size of the Pt, Pt-Mo, Pt-Zr, and Pt-Fe-Ir electrocatalysts was 3.75, 4.75, 4.29, and 3.3 nm. The composition ratio of the electrocatalyst was determined using EDX and the results are shown in Table 1. The respective atomic ratio of the Pt-Mo, Pt-Zr, and Pt-Fe-Ir electrocatalysts was 98:2, 80:20, and 80:10:10. This demonstrates that the alloy electrocatalyst for this study has a high content of Pt.

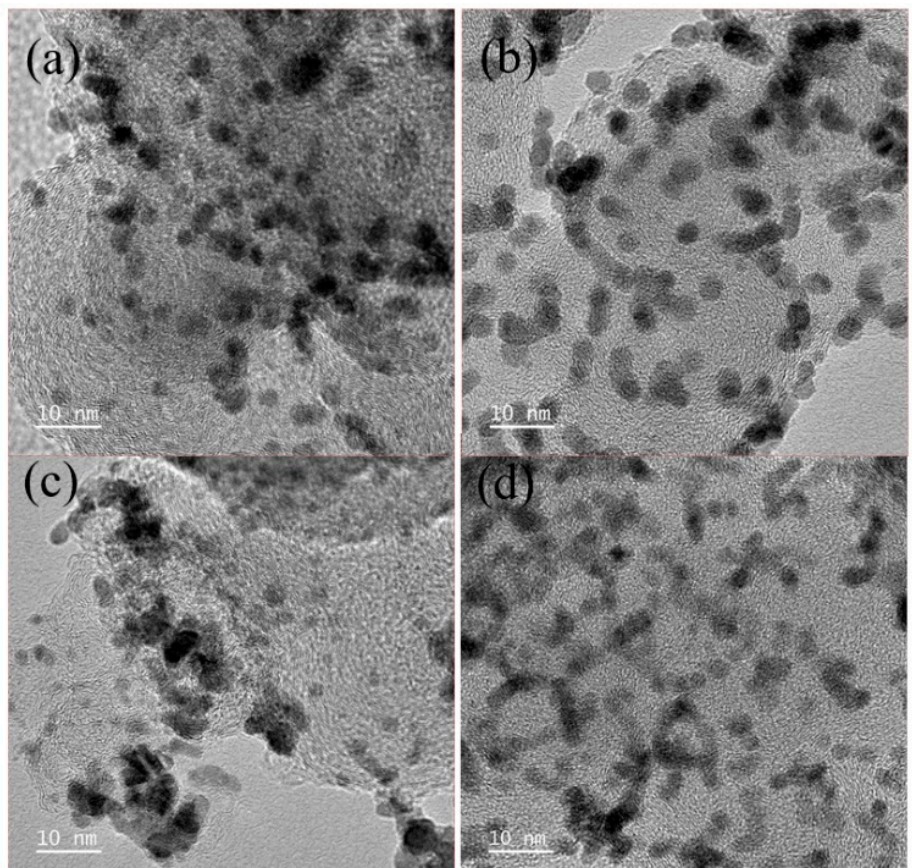

**Figure 1.** TEM images of the (**a**) Pt/C; (**b**) Pt-Mo/C; (**c**) Pt-Zr/C; and (**d**) Pt-Fe-Ir/C electrocatalysts.

**Table 1.** Composition and particle size for the Pt, Pt-Mo, Pt-Zr, and Pt-Fe-Ir electrocatalysts on carbon black support.

| Electrocatalysts | Composition (atm. %) | | | Average Particle Size (nm) |
|---|---|---|---|---|
| | Pt | $M_1$ | $M_2$ | |
| Pt | 100 | 0 | 0 | 3.75 |
| Pt-Mo | 98 | 2 | 0 | 4.57 |
| Pt-Zr | 80 | 20 | 0 | 4.29 |
| Pt-Fe-Ir | 80 | 10 | 10 | 3.3 |

The crystal structure of the Pt/C, Pt-Mo/C, Pt-Zr/C, and Pt-Fe-Ir/C electrocatalysts was determined using XRD. Figure 2 shows XRD patterns for the Pt/C, Pt-Mo/C, Pt-Zr/C, and Pt-Fe-Ir/C electrocatalysts. The diffraction peaks for the Pt/C electrocatalyst are located at 2θ angles of 39.89, 46.24, 67.42, and 81.53° and are attributed to the (111), (200), (220), and (311) planes of Pt face-centered cubic structure (fcc), respectively. Due to the different radii of different metal atoms, substitution of other atoms for the platinum atoms in the face-centered cubic structure results in slightly different bond distances. Position of the (111) peaks were slightly shifted, respectively, from 39.89° in the Pt/C to 39.86, 40.2, and 40.09° in the Pt-Mo/C, Pt-Zr/C, and Pt-Fe-Ir/C electrocatalysts, which demonstrates that the second or third metal atoms replaced some Pt atom positions to form alloy compounds.

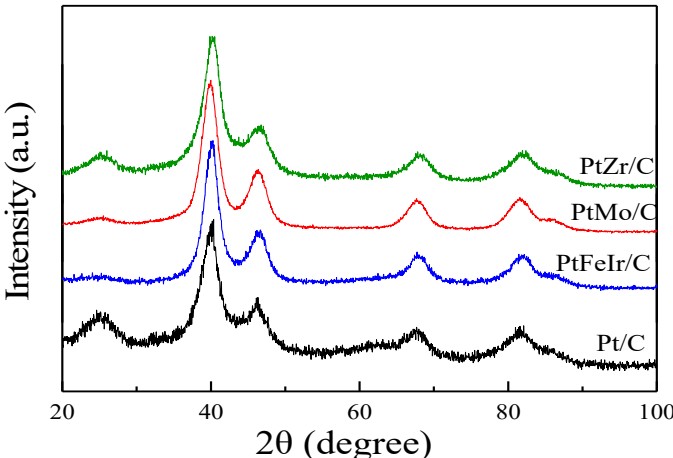

**Figure 2.** XRD patterns for the Pt-Mo/C, Pt-Zr/C, Pt-Fe-Ir/C, and Pt/C electrocatalysts.

## 2.2. Cyclic Voltammetry of Pt-Mo/C/GCE, Pt-Zr/C/GCE, Pt-Fe-Ir/C/GCE, and Pt/C/GCE

The electrochemical behavior of Pt/C/GCE, Pt-Mo/C/GCE, Pt-Zr/C/GCE, and Pt-Fe-Ir/C/GCE in the presence and absence of 25 mM AA in 0.5 M HClO$_4$ solution is shown in Figure 3. Figure 3a shows that the Pt/C/GCE displayed behavior that is typical for a Pt electrode in acid solution in the absence of 25 mM AA. Protons were adsorbed and desorbed between −0.2 and 0.15 V vs. Ag/AgCl. Pt began to form Pt oxide at 0.5 V vs. Ag/AgCl. In the presence of 25 mM AA, the increase in current from 0.1 to 0.8 V vs. Ag/AgCl and obvious oxidation wave appeared at 0.34 V vs. Ag/AgCl indicating AA oxidation on the surface of the Pt/C/GCE. The AA oxidation peak was also observed at 0.34 V vs. Ag/AgCl on Pt-Mo/C/GCE, Pt-Zr/C/GCE and Pt-Fe-Ir/C/GCE in Figure 3b–d. All electrodes could electrocatalyze AA oxidation. Among these electrodes, Pt-Mo/C/GCE had the highest AA oxidation peak current.

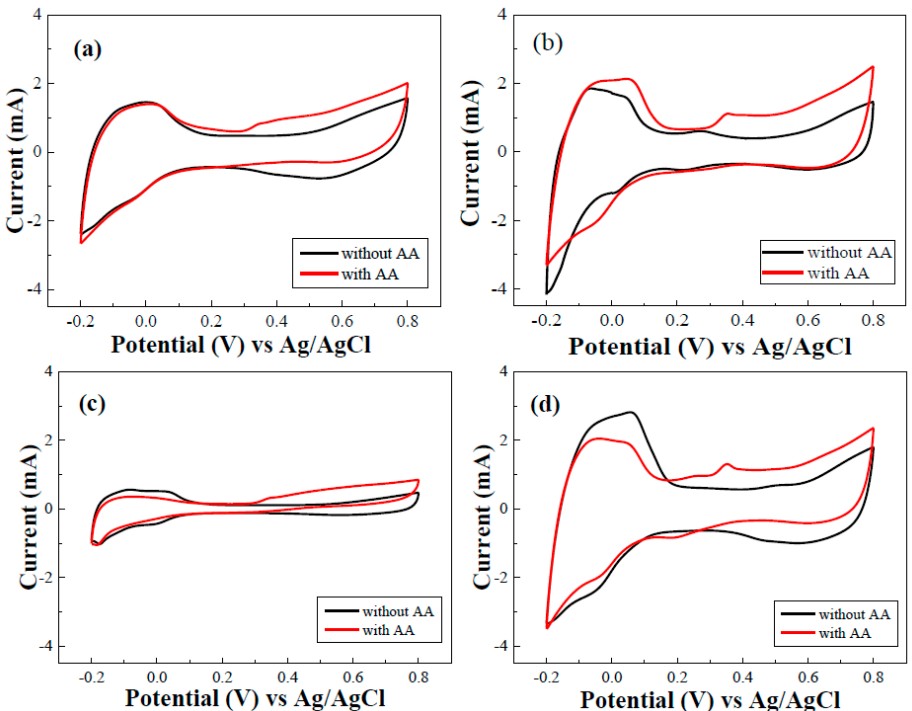

**Figure 3.** Cyclic voltammograms in the presence and absence of 25 mM AA for (**a**) Pt/C; (**b**) Pt-Mo/C; (**c**) Pt-Zr/C; and (**d**) Pt-Fe-Ir/C electrocatalysts: scan rate = 0.1 V/s, in 0.5 M HClO$_4$ solution.

CVs for Pt/C/GCE, Pt-Mo/C/GCE, Pt-Zr/C/GCE and Pt-Fe-Ir/C/GCE in the presence and absence of 10 mM DA in 0.5 M HClO$_4$ solution are shown in Figure 4. There was a pair of well-defined and almost reversible redox peaks at a potential of 0.4 to 0.7 V vs. Ag/AgCl for the four electrodes in the presence of 10 mM dopamine. The oxidation peak current was greater than that for an electrode that has no dopamine added, which demonstrates that the Pt/C/GCE, Pt-Mo/C/GCE, Pt-Zr/C/GCE, and Pt-Fe-Ir/C/GCE electro-catalyze dopamine oxidation.

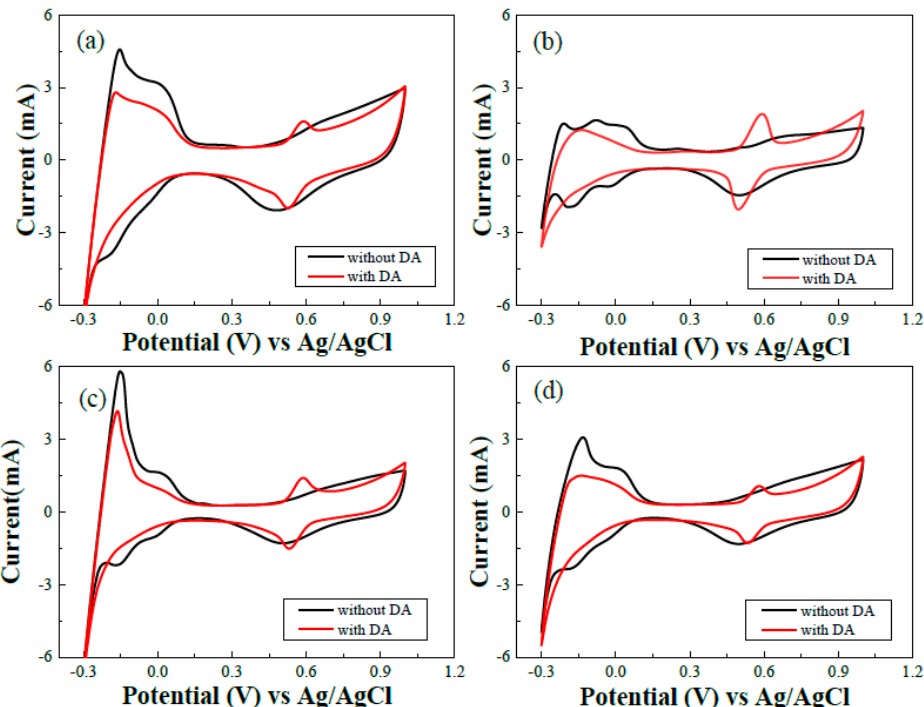

**Figure 4.** Cyclic voltammograms in the presence and absence of 10 mM DA for (**a**) Pt/C; (**b**) Pt-Mo/C; (**c**) Pt-Zr/C; and (**d**) Pt-Fe-Ir/C electrocatalysts: scan rate = 0.1 V/s, in 0.5 M HClO$_4$ solution.

The electroactive surface area of the electrodes was calculated using the Randles–Sevcik equation [25]. The relationship between current and scan rate was determined using cyclic voltammetry in a 5 mM K$_3$[Fe(CN)$_6$] using a 0.1 M KCl supporting electrolyte, as described in the following:

$$I_p = 2.69 \times 105 \, n^{3/2} A \, D^{1/2} C v^{1/2} \tag{1}$$

$$R_f = A / A_{geom} \tag{2}$$

where I$_p$, n, A, D, C, and v are the anodic peak current, the number of electrons (1 for K$_3$[Fe(CN)$_6$]) involved in the reaction, electroactive surface area (cm$^2$), the electroactive surface area (cm$^2$), the diffusion coefficient (7.6 × 10$^{-6}$ cm$^2$ s$^{-1}$ for K$_3$[Fe(CN)$_6$]), the concentration of the reactant (mol cm$^{-3}$), and the scan rate (V s$^{-1}$), respectively. Equations (1) and (2) are used to calculate the electroactive surface area and the roughness factor for different electrodes, as shown in Table 2. The Pt-Mo/C/GCE has a large electroactive surface area and a large roughness factor; therefore, more catalytic active sites are available on the electrode surface, which accelerates electron transfer and provides more efficient electrochemical sensing.

### 2.3. Determination of Appropriate Sensing Potential

The polarization curve was used to determine the optimal oxidation potential of the electrocatalytic electrode for sensing vitamin C and dopamine. A stable current value was established at each potential using chronoamperometry at different potentials. The polarization curve was obtained by plotting the potential and the steady current. Figure 5

shows the respective polarization curves for the Pt/C/GCE, Pt-Mo/C/GCE, Pt-Zr/C/GCE, and Pt-Fe-Ir/C/GCE, measured in 15 mM AA and 10 mM DA solutions. The net current value was obtained by subtracting the background current from the current in the analyte.

**Table 2.** Comparison of electroactive area and roughness factor for various electrodes.

| Electrode | Anodic Peak Current (μA) | Electroactive Area (cm$^2$) | Roughness Factor |
|---|---|---|---|
| GCE | 32 | 0.039 | 0.55 |
| Pt/C/GCE | 173 | 0.161 | 2.27 |
| Pt-Mo/C/GCE | 181 | 0.176 | 2.49 |
| Pt-Zr/C/GCE | 81 | 0.113 | 1.17 |
| Pt-Fe-Ir/C/GCE | 103 | 0.083 | 1.61 |

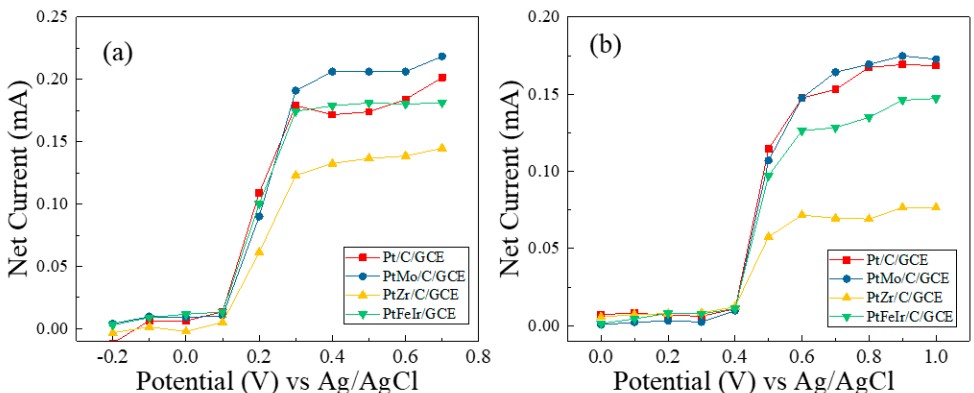

**Figure 5.** The polarization curves that were obtained by subtracting background currents for (**a**) 15 mM AA; (**b**) 10 mM DA for the Pt/C/GCE, Pt-Mo/C/GCE, Pt-Zr/C/GCE, and Pt-Fe-Ir/C/GCE.

Figure 5a shows that the Pt/C/GCE, Pt-Mo/C/GCE, Pt-Zr/C/GCE, and Pt-Fe-Ir/C/GCE did not electrocatalyze AA oxidation at potentials between −0.2 and 0.1 V vs. Ag/AgCl. If the potential was increased from 0.1 to 0.3 V vs. Ag/AgCl, the response current increased as the potential increased, which shows that this region was kinetically controlled and not suitable for sensing AA. If the potential was increased from 0.3 to 0.7 V vs. Ag/AgCl, the response current for catalytic AA oxidation reached the limiting current plateau and this region was controlled by mass transfer. Therefore, 0.3 V vs. Ag/AgCl was the potential that was used for subsequent sensing of AA for this study. The Pt-Mo/C/GCE also had a higher steady-state response current than other electrodes.

Figure 5b shows the polarization curves in 10 mM DA. The Pt/C/GCE, Pt-Mo/C/GCE, Pt-Zr/C/GCE, and Pt-Fe-Ir/C/GCE had no electrocatalytic effect on DA between a potential of 0 and 0.4 V vs. Ag/AgCl. If the potential was increased from 0.4 to 0.6 V vs. Ag/AgCl, the response current increased rapidly as the potential was increased, which demonstrates that this is a kinetically controlled region. The limiting current plateaus showed at a potential between approximately 0.6 and 1.0 V vs. Ag/AgCl. This limiting current plateau region was controlled by mass transfer; therefore, 0.6 V vs. Ag/AgCl was the applied potential for sensing DA and the response currents for this potential to DA sensing was recorded. The Pt-Mo/C/GCE also had a higher steady-state response current to DA oxidation than other electrodes.

### 2.4. Sensing Performances of Pt-Mo/C/GCE, Pt-Zr/C/GCE, Pt-Fe-Ir/C/GCE, and Pt/C/GCE

Figure 6a shows the amperometric response curves for Pt/C/GCE, Pt-Mo/C/GCE, Pt-Zr/C/GCE, and Pt-Fe-Ir/C/GCE in 0.5 M HClO$_4$ solution at a sensing potential of 0.3 V for various concentrations of AA. When the background current had stabilized, AA solution was added to achieve a concentration of AA of 1.5 mM. When the current value had stabilized, AA solution was added to achieve a concentration of AA of 3 mM. Adding

AA solution to increase the AA concentration from 1.5 mM to 15 mM produced a stepwise increase in the current response curve.

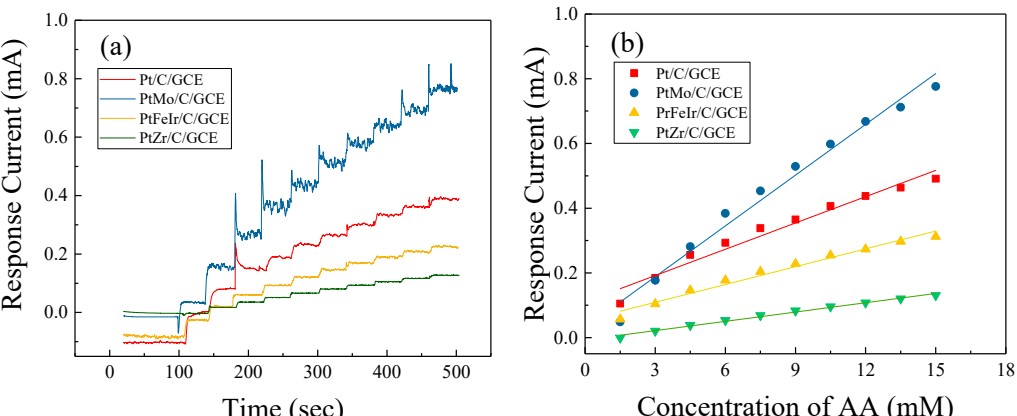

**Figure 6.** (**a**) Amperometric response; and (**b**) calibration curves for Pt/C/GCE, Pt-Mo/C/GCE, Pt-Zr/C/GCE and Pt-Fe-Ir/C/GCE for various concentrations of AA from 1.5 to 15 mM at an applied potential of 0.3 V vs. Ag/AgCl.

The Pt-Mo/C/GCE performed best in sensing AA. However, as the AA concentration increased, noise also increased, possibly because the products of AA oxidation were also adsorbed onto the electrode surface, which increased surface noise. The concentration calibration curve was obtained by plotting the response current against the AA concentration, as shown in Figure 6b. The sensing response current had a linear relationship with the AA concentration. The gradient of the plot was the sensitivity of the catalyst electrode to AA. The greater the gradient, the greater was the sensitivity of the electrode. The respective sensitivity for the Pt/C/GCE, Pt-Mo/C/GCE, Pt-Zr/C/GCE, and Pt-Fe-Ir/C/GCE at a sensing concentration of 1.5 to 15 mM AA was 27.09 ($R^2$ = 0.962), 31.29 ($R^2$ = 0.979), 9.57 ($R^2$ = 0.973), and 18.36 ($R^2$ = 0.990) $\mu$A mM$^{-1}$.

Figure 7a shows the amperometric response for a Pt-Mo/C electrode for various concentrations of DA from 1 to 10 mM at an applied potential of 0.6 V vs. Ag/AgCl. When the background current had stabilized, 1 to 10 mM dopamine was added. As the concentration of dopamine increased, the response current increased significantly. However, the noise also increased as the DA concentration increased, possibly because dopamine oxide that was adsorbed onto the electrode surface increased the noise. The concentration calibration curve is a plot of the response current against the DA concentration, as shown in Figure 7b. There was a linear relationship between dopamine concentration and response current.

A comparison of the DA sensitivity for Pt/C/GCE, Pt-Mo/C/GCE, Pt-Zr/C/GCE, and Pt-Fe-Ir/C/GCE at constant potentials of 0.6 V vs. Ag/AgCl shows that the Pt-Mo/C/GCE had the highest sensitivity to DA. The sensitivity of the Pt-Mo/C/GCE to DA was 72.24 ($R^2$ = 0.993) $\mu$A mM$^{-1}$ (1021 $\mu$M mM$^{-1}$ cm$^{-2}$) at 0.6 V vs. Ag/AgCl. The sensitivity for the Pt/C/GCE, Pt-Zr/C/GCE, and Pt-Fe-Ir/C/GCE at a sensing concentration of 1 to 10 mM DA was 68.23 ($R^2$ = 0.995), 47.76 ($R^2$ = 0.995), and 64.29 ($R^2$ = 0.994) $\mu$A mM$^{-1}$.

Differential pulse voltammetry (DPV) was also used to determine the sensitivity of Pt-Mo/C/GCE for DA because it is highly sensitive. Figure 7c shows the DPV curves for the Pt-Mo/C/GCE for DA concentrations of 1 to 10 mM in the solutions. The oxidation peak for DA for the Pt-Mo/C/GCE in DPV was at 0.56 V vs. Ag/AgCl. Increasing the DA concentration produced a series of continuously increasing currents. Figure 7d shows that the peak current was proportional to the DA concentration from 1 to 10 mM. The sensitivity of the Pt-Mo/C/GCE to DA in DPV was 152.55 ($R^2$ = 0.974) $\mu$A $\mu$M$^{-1}$ (2157.7 $\mu$A $\mu$M$^{-1}$ cm$^{-2}$). The detection limit was calculated to be 24 $\mu$M.

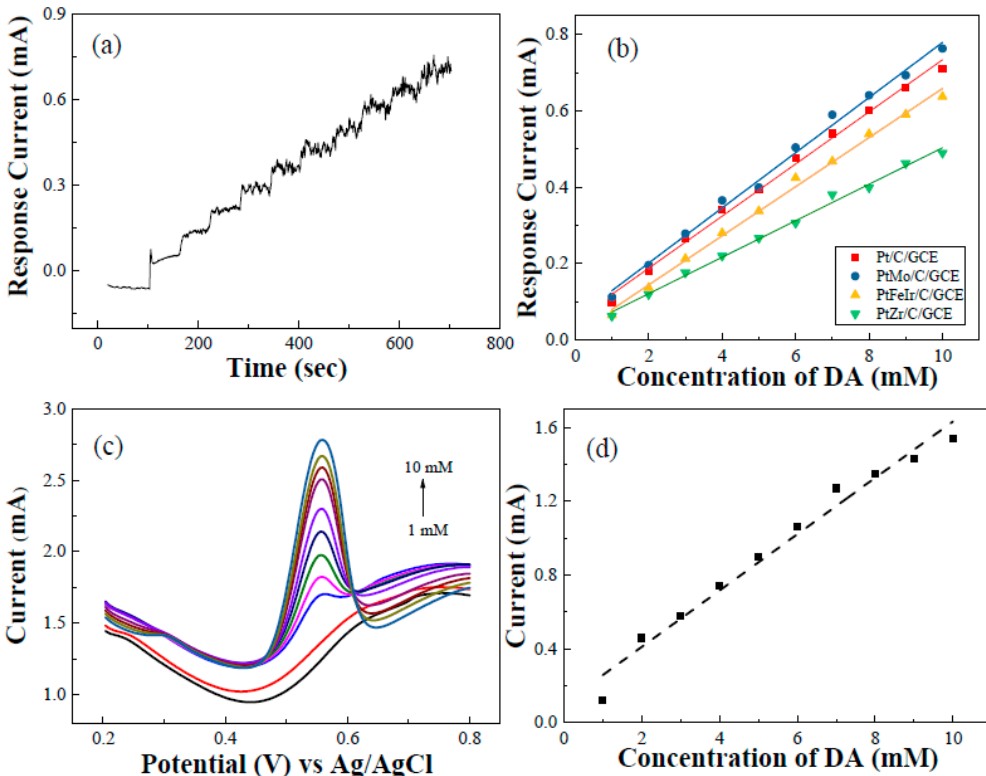

**Figure 7.** (**a**) Amperometric response for Pt-Mo/C/GCE for concentrations of DA from 1 to 10 mM at an applied potential of 0.6 V vs. Ag/AgCl; (**b**) calibration curves for Pt/C/GCE, Pt-Mo/C/GCE, Pt-Zr/C/GCE, and Pt-Fe-Ir/C/GCE for concentrations of DA from 1 to 10 mM at an applied potential of 0.6 V vs. Ag/AgCl; (**c**) DPV curves for Pt-Mo/C/GCE for concentrations of DA from 1 to 10 mM; and (**d**) calibration curve for Pt-Mo/C/GCE from (**c**).

Figure 8a,b respectively show the response time for the Pt/C/GCE, Pt-Mo/C/GCE, Pt-Zr/C/GCE, and Pt-Fe-Ir/C/GCE for the detection of AA and DA. The response time is defined as the time that is required for the sensing current to reach 90% of the stable current value. The average response time for these four electrodes for sensing AA was between 4 and 6 s and the average response time that was required to sense DA was between 4 and 8 s. The Pt-Mo/C/GCE had the shortest response time and measured the concentration of the analyte assured in 4 s.

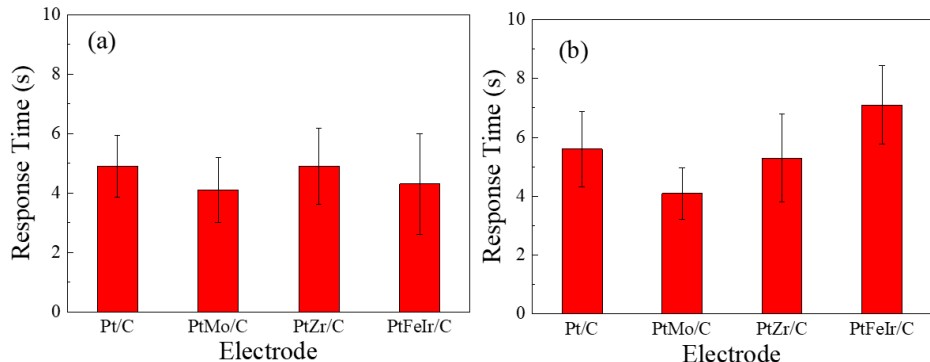

**Figure 8.** Response time for the detection of (**a**) AA; and (**b**) DA for the Pt/C/GCE, Pt-Mo/C/GCE, Pt-Zr/C/GCE and Pt-Fe-Ir/C/GCE.

The Pt-Mo/C/GCE exhibited excellent selectivity. Sucrose, citric acid, tartaric acid, and uric acid were selected as interferents to determine the selectivity of the sensing electrodes. Figure 9a shows the response current for the Pt-Mo/C/GCE electrode at 0.3 V vs. Ag/AgCl.

Sucrose, citric acid, tartaric acid, uric acid, and AA were added at a concentration of 0.5 mM. The current values for these interfering substances were negligible, and after adding 0.5 mM AA, the current values demonstrated a significant change.

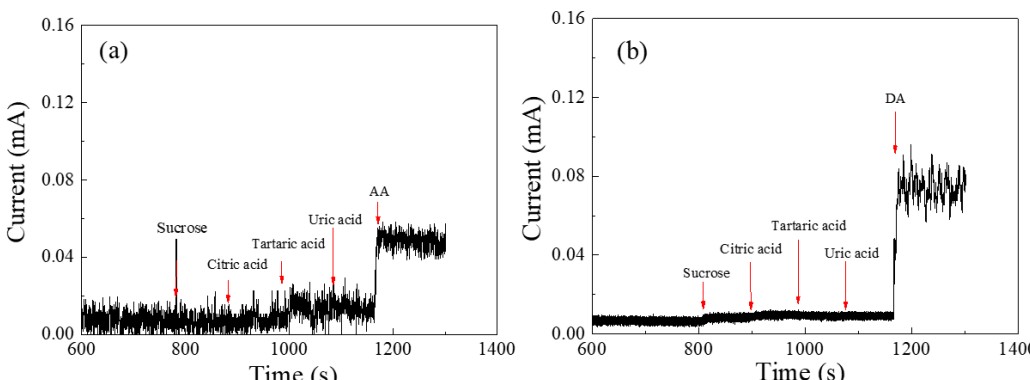

**Figure 9.** Selectivity of the Pt-Mo/C/GCE for the detection of (**a**) AA; and (**b**) DA in the presence of sucrose, citric acid, tartaric acid, and uric acid at a concentration of 0.5 mM.

Figure 9b shows the response current for the Pt-Mo/C/GCE at 0.6 V vs. Ag/AgCl when sucrose, citric acid, tartaric acid, uric acid, and DA were added at a concentration of 0.5 mM. The current values for sucrose, citric acid, tartaric acid, and uric acid were almost unchanged, but the addition of 0.5 mM DA significantly increased the current values, which demonstrates that Pt-Mo/C/GCE is highly selective to AA and DA. Other electrodes demonstrated similar results to those for the Pt-Mo/C/GCE.

The reproducibility of Pt-Mo/C/GCE was determined using five independent Pt-Mo/C/GCEs for sensing AA and DA and the results are shown in Figure 10a. The sensitivity of each electrode divided by the sensitivity that is measured by the first electrode is defined as the normalized sensitivity. The results show that Pt-Mo/C/GCE had acceptable sensitivity, and the respective standard deviations for detecting AA and DA were 7.07% and 5.45%.

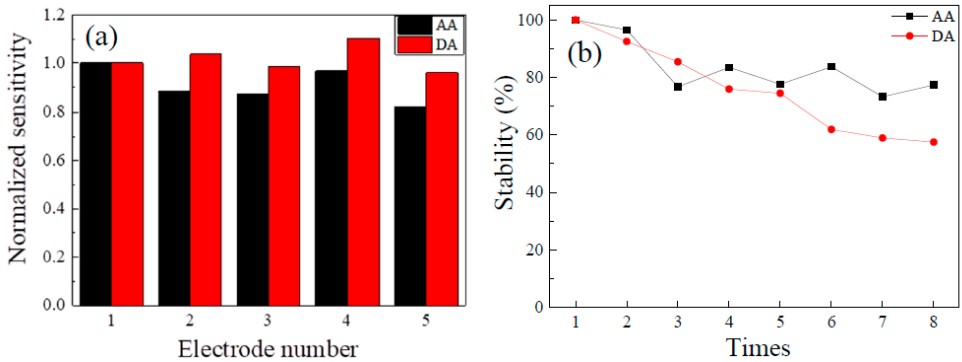

**Figure 10.** (**a**) Repeatability of the Pt-Mo/C/GCE for the detection of AA and DA; (**b**) stability of the Pt-Mo/C/GCE for the detection of AA and DA.

Figure 10b shows the results for the stability test for the Pt-Mo/C/GCE for sensing AA and DA. The electrodes were tested a total of eight times on different days. In terms of the AA-sensing part, the sensitivity of the Pt-Mo/C/GCE decreased initially. After the third test, the sensitivity remained at around 80%. In terms of the DA-sensing part, the sensitivity of the Pt-Mo/C/GCE decreased as the number of tests increased. During the 8th test, the electrode sensitivity decreased to 57%; therefore, there was an opportunity for improvement.

Cyclic voltammograms for the Pt-Mo/C/GCE for various scan rates (25–150 mV s$^{-1}$) in the presence of 15 mM AA and 10 mM DA in 0.5 M HClO$_4$ solution are shown in Figure 11a,b. The Pt-Mo/C electrocatalyst had a significant catalytic effect at these scan rates. The relationship between the anodic peak current for catalytic oxidation of the

analyte (15 mM AA and 10 mM DA, respectively) and the square root of the scan rate is shown in Figure 11c,d. The anodic peak current for AA and DA increased linearly with the square root of the scan rate for rates of less than 150 mV s$^{-1}$, which shows that the reaction was controlled by mass transfer, according to Equation (3) [26]:

$$i_{pa} = 0.496nFACD^{1/2}F^{1/2}\nu^{1/2}R^{-1/2}T^{-1/2} \tag{3}$$

where A is the geometric area of the glassy carbon electrode and C and D are the concentration and diffusion coefficient of the bulk species, respectively. Figure 11c,d, and Equation (3) were used to calculate the diffusion coefficient for AA and DA (D = 2.42 × 10$^5$ cm$^2$ s$^{-1}$ and 5.23 × 10$^5$ cm$^2$ s$^{-1}$, respectively) in 0.5 M HClO$_4$ solution (pH 0.28 at 25 °C).

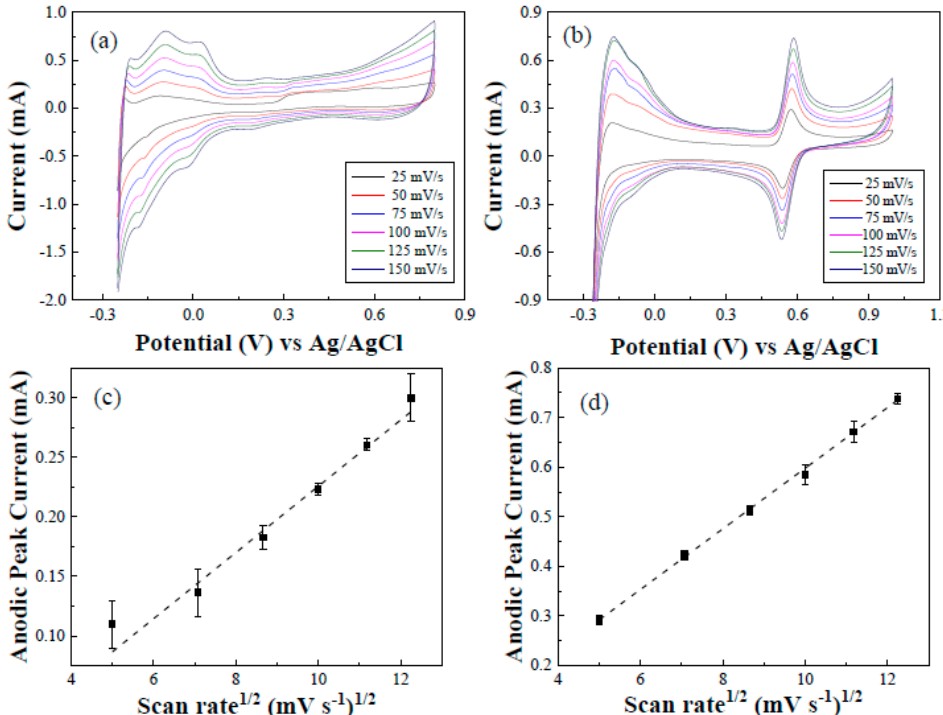

**Figure 11.** Cyclic voltammograms for the Pt-Mo/C/GCE in 0.5 M HClO$_4$ solution for scan rates of 25, 50, 75, 100, 125, and 150 mV s$^{-1}$ in the presence of (**a**) 15 mM AA; and (**b**) 10 mM DA and anodic peak current versus the square root of the potential scan rate in the presence of (**c**) 15 mM AA; and (**d**) 10 mM DA.

Chronoamperometry was used to determine the catalytic rate constant (k). The catalytic rate constant for the oxidation of AA and DA at the Pt-Mo/C/GCE was calculated using the relationship [27,28] (Equation (4)):

Chronoamperometry was employed for the evaluation of the catalytic rate constant (k). The catalytic rate constant for the oxidation of AA and DA at the Pt-Mo/C/GCE was estimated from the relationship [27,28] (Equation (4)):

$$I_{cat}/I_L = \pi^{1/2}(kCt)^{1/2} \tag{4}$$

where I$_{cat}$ and I$_L$ are the respective currents in the presence and absence of AA or DA, k is the catalytic rate constant, and t is the time in seconds. A chronoamperometric test used an operating condition at a potential of 0.3 V vs. Ag/AgCl for AA and 0.7 V vs. Ag/AgCl for DA. The gradient of the I$_{cat}$/I$_L$ versus t$^{1/2}$ plot is used to calculate k for a specific concentration of analyte. Figure 12a,b show the relationship between I$_{cat}$/I$_L$ and t$^{1/2}$ using the data for chronoamperometry. The value of k is 1.2 × 10$^2$ and 2.23 × 10$^2$ M$^{-1}$ s$^{-1}$ for AA and DA, respectively, on Pt-Mo/C/GCE.

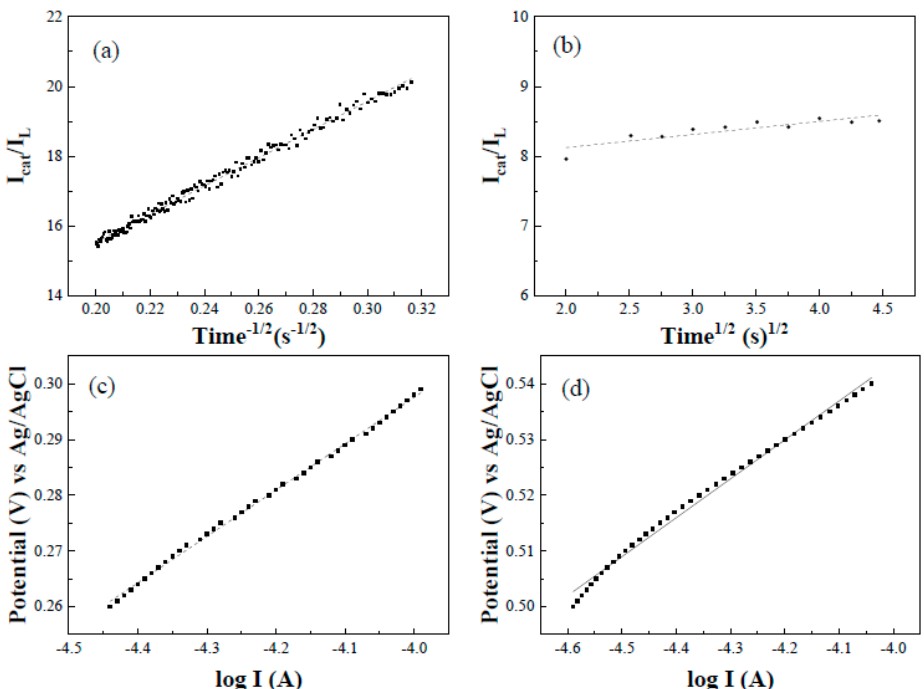

**Figure 12.** Dependence of $I_{cat}/I_L$ on $t^{1/2}$ derived from the data of chronoamperometry in the presence of (**a**) 9 mM AA; and (**b**) 10 mM DA. Tafel plot of the Pt-Mo/C/GCE in the presence of (**c**) 10 mM AA; and (**d**) 15 mM DA.

To calculate the heterogeneous standard rate constant ($k^o$), the Tafel (log I vs. E) behavior of the Pt-Mo/C/GCE in the presence of 10 mM AA and 15 mM DA was determined, as shown in Figure 12c,d, respectively. In this plot, the value of current and potential were calculated using the increase in the oxidation peak for the CV. This plot has a respective gradient of 83 mV s$^{-1}$ and 69 mV s$^{-1}$ for AA and DA. The Tafel equation and the relation between exchange current ($i^o$) and $k^o$ were used to calculate $k^o$ as $6.19 \times 10^{-3}$ and $7.66 \times 10^{-3}$ cm s$^{-1}$ for AA and DA, respectively, on Pt-Mo/C/GCE.

Table 3 compares the sensitivity and response time of the Pt/C/GCE, Pt-Mo/C/GCE, Pt-Zr/C/GCE, and Pt-Fe-Ir/C/GCE for AA and DA oxidation. Among the electrodes discussed, Pt-Mo/C/GCE had the highest sensitivity and the shortest response time for sensing AA or DA. The high sensing performance of this electrode comes from its high electroactive area and high catalytic rate constant.

**Table 3.** Comparison of sensitivity and response time for various electrodes.

| Electrode | Sensitivity ($\mu A$ mM$^{-1}$) | | Response Time (s) | |
|---|---|---|---|---|
| | **AA** | **DA** | **AA** | **DA** |
| Pt/C/GCE | 27.09 | 68.23 | 4.9 | 5.6 |
| Pt-Mo/C/GCE | 31.29 | 72.24 | 4.1 | 4.1 |
| Pt-Zr/C/GCE | 9.57 | 47.76 | 4.9 | 5.3 |
| Pt-Fe-Ir/C/GCE | 18.36 | 64.29 | 4.3 | 7.1 |

## 3. Materials and Methods

### 3.1. Reagents

All reagents, including chloroplatinic acid hexahydrate ($H_2PtCl_6 \cdot 6H_2O$), molybdenum pentachloride ($MoCl_5$), zirconium tetrachloride ($ZrCl_4$), iron (III) nitrate nonahydrate ($Fe(NO_3)_3 \cdot 9H_2O$), iridium(IV) chloride hydrate ($IrCl_4 \cdot xH_2O$), ethylene glycol ($C_2H_6O_2$), sodium borohydride ($NaBH_4$), hydrochloric acid (HCl), perchloric acid ($HClO_4$), Vulcan XC-72 carbon black, AA, DA, and interferents, were purchased from commercial sources.

The reagents are of analytical grade and were used without further purification. A stock solution of AA and DA was prepared daily using deionized (DI) water.

### 3.2. Preparation of Pt/C, Pt-Mo/C, Pt-Zr/C, and Pt-Fe-Ir/C Electrocatalysts

DI water was used as the solvent to prepare $Pt^{4+}$, $Mo^{5+}$, $Zr^{4+}$, $Fe^{3+}$, and $Ir^{4+}$ precursor solutions with a concentration of 0.1 M. The metal precursor solution (5 mL) was produced to a specified ratio. After stirring for 20 min, 0.1 g of XC-72 carbon powder was added and the mixture was stirred again for 20 min. A volume of 10 mL of ethylene glycol was added, and the mixture was stirred using ultrasonic vibration for 1 h; 0.2 g of sodium borohydride was added, and the mixture was stirred for 4 h. Next, 10% 10 mL aqueous HCl was added and the mixture was stirred for 10 min. A vacuum pump was used to filter the electrocatalyst.

The electrocatalyst was rinsed several times with DI water, and then the electrocatalyst was dried in a vacuum oven at 80 °C for 24 h. The dried electrocatalyst powder was mixed with 5 wt% Nafion® solution and DI water to produce an electrocatalyst slurry with a weight percentage of 1:9.25:8.75.

### 3.3. Preparation of Pt/C/GCE, Pt-Mo/C/GCE, Pt-Zr/C/GCE, and Pt-Fe-Ir/C/GCE

The GCE was first polished with 1000-grit sandpaper and then rinsed with DI water. It was then polished with 1500-grit sandpaper and washed with DI water again. A 10 μL sample of Pt/C, Pt-Mo/C, Pt-Zr/C, or Pt-Fe-Ir/C slurry was injected onto the bare GCE surface and then maintained at 60 °C for 10 min to produce the Pt/C/GCE, Pt-Mo/C/GCE, Pt-Zr/C/GCE, or Pt-Fe-Ir/C/GCE. The geometric area of the GCE was 0.0707 $cm^2$.

### 3.4. Characterization of Pt/C, Pt-Mo/C, Pt-Zr/C, and Pt-Fe-Ir/C Electrocatalysts

The surface morphology of the Pt/C, Pt-Mo/C, Pt-Zr/C, and Pt-Fe-Ir/C electrocatalysts and their quantitative measurement was determined using a HR-TEM (JEM-2100Plus) equipped with an EDS. The structure of these four electrocatalysts was determined using an XRD (D8 SSS, Bruker Co., Billerica, MA, USA). A potentiostat (VersaSTAR 3, METEK Co., Oak Ridge, TN, USA) was used to determine the efficacy with which the electrocatalysts oxidize AA and DA. Electrochemical experiments were performed using a three-electrode system, with a Pt wire and Ag/AgCl electrodes as counter and reference electrodes, respectively. The electrolyte was 0.5 M $HClO_4$ solution.

## 4. Conclusions

Detection of AA and DA is important for the diagnosis of health conditions in humans and for improving the quality of living. Four modified electrodes, a Pt-Mo/C/GCE, Pt-Zr/C/GCE, Pt-Fe-Ir/C/GCE, and a Pt/C/GCE, were fabricated and used for the amperometric detection of AA and DA. Compared to the Pt/C/GCE, the Pt-Zr/C/GCE and Pt-Fe-Ir/C/GCE, Pt-Mo/C/GCE were highly sensitive to AA and DA at different applied potentials. The sensitivity of the Pt-Mo/C/GCE was 31.29 μA $mM^{-1}$ at 0.3 V vs. Ag/AgCl and 72.24 μA $mM^{-1}$ at 0.6 V vs. Ag/AgCl for AA and DA, respectively; therefore, the Pt-Mo/C/GCE is an electrocatalyst that is highly effective for AA and DA detection. The respective limits of detection for AA and DA were 7.69 and 6.14 μM. The Pt-Mo/C/GCE demonstrated acceptable reproducibility and stability. This electrode was also less sensitive to interfering substances, such as sucrose, citric acid, tartaric acid, uric acid, than to AA or DA. To assist in point-of-care testing, a portable sensor constructed by depositing Pt-Mo/C on a paper substrate could be prepared in the future.

**Author Contributions:** Conceptualization, Y.-C.W.; methodology, J.-Y.S.-C., T.-Y.Y. and C.-L.C.; investigation, J.-Y.S.-C., T.-Y.Y. and C.-L.C.; writing—original draft preparation, Y.-C.W.; writing—review and editing, Y.-C.W.; supervision, Y.-C.W.; project administration, Y.-C.W. All authors have read and agreed to the published version of the manuscript.

**Funding:** This research was funded by National Science and Technology Council, grant number MOST109-2221-E-035-024-MY3.

**Institutional Review Board Statement:** Not applicable.

**Informed Consent Statement:** Not applicable.

**Data Availability Statement:** Not applicable.

**Acknowledgments:** The authors gratefully acknowledge support from the National Science and Technology Council (MOST109-2221-E-035-024-MY3) and from Feng Chia University. The authors also appreciate the help of the Precision Instrument Support Center of Feng Chia University in providing measurement facilities.

**Conflicts of Interest:** The authors declare no conflict of interest.

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
