# Peer review of "A Comparison of the Sensing Behavior for Pt-Mo/C-, Pt-Zr/C-, Pt-Fe-Ir/C-, and Pt/C-Modified Glassy Carbon Electrodes for the Oxidation of Ascorbic Acid and Dopamine"

_catalysts, doi:10.3390/catal13020337_

Round 1

Reviewer 1 Report

In the paper entitled "A Comparison of the Sensing Behavior for Pt-Mo/C, Pt-Zr/C, Pt-Fe-Ir/C and Pt/C Modified Glassy Carbon Electrodes for the Oxidation of Ascorbic Acid and Dopamine", Weng and co-workers presented the fabrication, characterization and potential utilization of four sensing materials for the determination of AA and DA. Although the obtained results are very interesting, the manuscript has to be revised after major revision. Some issues should be corrected:

1. The Introduction: please add some paragraphs describing the use of Pt-based sensing materials in the determination of other analytes - it should be emphasized why did the Authors choose this type of material.

2. Please remove the last sentence from the Intro - it already describes the obtained results, whereas it is only an introduction to the whole story.

3. Why did the Authors use the exact conc. of metal salts? Can you modulate the content of Pt in your materials? In my opinion, diverse Pt content should be assessed.

4. Lines 112-118 and Fig. 3a-d - There is no clear oxidation peak of AA in the investigated potential window, whereas CV scans of DA demonstrated a well-revealed oxidation peak at approx. 0.6 V. How can you be sure that the AA is oxidized at 0.6 V? The current increase from 0.6 to 0.8 V is also present in CV voltammograms without AA. Please explain.

5. Figure 5b - if the polarization curve is still growing between 0.5 and 0.8 V for Pt/Mo/C/GCE, the applied potential should be increased to the higher values to obtain the stable plateau.

6. Please add the R2 values for plots in Fig. 6b and 7b.

7. Please explain why did you choose the conc. of the interferences of 0.5 mM. Why not higher?

Reviewer 2 Report

In this manuscript, several Pt-base alloys were supported on carbon to modify GCE as sensors for sensitively determining AA or DA. From various electrochemical characterization means, Pt-Mo system was found to exhibit good sensitivity, LOD, and reproducibility. Also, some kinetic constants were determined electrochemically. The obtained results laid solid foundation for the possible acceptance of the manuscript. Therefore, the reviewer recommends a major revision before its acceptance. The detailed comments are as follows:

1.       As is stated in Introduction section, “Few studies concern the application of…”, and “… has not been demonstrated”. In fact, these are not the reason for the authors to carry out the research. Why these alloys are selected for the electrocatalysis towards AA and DA? Maybe the importance or the necessity is the true reason for the research;

2.       When discussing Figure 2, the authors are suggested to give the exact data to convince the readers that the peaks are really slightly shifted, and to give further explanation of the reason;

3.       No oxidation peak is observed in the CV curves of AA, why?

4.       For all the comparisons, Pt-Mo system gives the best performance. However, all the comparisons only give the result, without any discussion on the relationship between catalyst structure and electrochemical activity. Besides, there is a lack of structural and morphological characterization of the catalysts;

5.       Differential pulse voltammetry was used in the case of DA, is it applicable to AA?

6.       Expression details: RGO and rGO should be unified; “The sensitivity, response time, selectivity and stability of … is determined” should be “… are determined”; “Pr-Mo” should be “Pt-Mo”; “Figure 5(a) shows that … does not…” should be “… do not…”; The slope values of 27.09, 31.29, 9.57, and 18.36 for AA should be carefully checked according to the curves in Figure 6(b); Figure 7(c) shows a concentration range of 0–1000 µM, not 1–10 mM as claimed in its legend; Eq. 3 is wrongly noted as Eq. 1 in the text.

Reviewer 3 Report

1.    It is not clear the role of each material on the overall sensing performance of the nanosensor. Please specify this clearly.

2.    To discuss point of care diagnosis in the conclusions, please discuss how to translate the sensing strategy into portable devices.

3.    From Fig. 11, please add error bars

4.    Fabrication and testing of the sensing electrode". Why did use the traditional drop-casted technology for the modification of electrodes? What is the binding force between the material and the electrode?

5.       The literature review is very poor. Some recent studies are missing. Papers related to paper subjects have appeared recently and should be cited.  https://doi.org/10.3390/catal12121528, https://doi.org/10.1016/j.surfin.2022.102455

Round 2

Reviewer 1 Report

Dear Authors,

All my comments have been addressed. The paper can be published in its present form.

Reviewer 2 Report

The authors have responsed the concersn from the reviewers and made their manuscript acceptable at its current status.